# End-Of-Day LED Lightings Influence the Leaf Color, Growth and Phytochemicals in Two Cultivars of Lettuce

**Yamin Li**[ID]**, Rui Shi, Haozhao Jiang, Linyuan Wu, Yiting Zhang, Shiwei Song**[ID]**, Wei Su and Houcheng Liu \***

College of Horticulture, South China Agricultural University, Guangzhou 510642, China;
yaminli@stu.scau.edu.cn (Y.L.); ruishi27@stu.scau.edu.cn (R.S.); jhzh111@stu.scau.edu.cn (H.J.);
linyuanwu@stu.scau.edu.cn (L.W.); yitingzhang@scau.edu.cn (Y.Z.); swsong@scau.edu.cn (S.S.);
susan_l@scau.edu.cn (W.S.)

\* Correspondence: liuhch@scau.edu.cn; Tel.: +86-020-85280464

**Abstract:** Four light treatments (W: white light; EOD-B: end-of-day enhanced blue light; EOD-FR: end-of-day supplementary far-red light; EOD-UV: end-of-day supplementary ultraviolet-A light) were designed to explore the effects of end-of-day (EOD) lightings (30 min before dark period) on leaf color, biomass and phytochemicals accumulation in two lettuce cultivars (*Lactuca sativa* cv. 'Red butter' and 'Green butter') in artificial light plant factory. EOD-FR stimulated the plant and shoot biomass of two cultivars, and EOD-B suppressed the growth of 'Red butter' but induced higher biomass in 'Green butter'. EOD lightings generated brighter, greener and yellower leaf in 'Red butter' at harvest, but the highest lightness and the deepest redness of 'Green butter' leaf were observed in the middle growth stage. 'Red butter' had prominent higher contents of chlorophylls and carotenoids, while these pigments showed less sensitivity to the interaction of cultivars and EOD lightings. EOD lightings impeded the accumulation of anthocyanin in two cultivars, except EOD-UV slightly increased the anthocyanin contents in 'Green butter'. EOD-UV strengthened the antioxidant capability of 'Green butter', but EOD lightings had different effects on the antioxidant and nutritional compound contents in two lettuce cultivars.

**Keywords:** end-of-day lighting; blue light; far-red light; *Latuca sativa*; phytochemicals; ultraviolet-A

## 1. Introduction

The increasing world population combined with decreasing arable land provide an excellent opportunity for the development of plant factory in urban areas. In addition, the aroused concern of food safety requires precision in environmental control to balance the yield production, agronomic characteristics, and the nutritional qualities of vegetable crops. Lettuce (*Latuca sativa* L.) is largely consumed in the world, and it is arguably one of the most common crops in plant factories with artificial light (PFALs). Lettuce is rich in natural pigments like anthocyanins, carotenoids and chlorophylls, and nutritional compounds such as vitamin C, proteins, and phenolics [1,2]. These phytochemicals not only contribute to the leaf color and/or the flavor of lettuce plant, but also benefit human health.

Light functions as the energy source to drive photosynthesis and as a signal that directly or indirectly regulates the plant morphology and physiology [3–5]. Plants perceive light via a complex array of photoreceptors (phytochromes, cryptochromes, phototropins and UVR8), which are defined by the absorbed light wavelength. When the light environment (light quality, quantity, directionality, and photoperiod) changes, these photoreceptors can switch forms or be activated in distinct manners [6,7]. Consequently, they transduce diverse light signals to modulate the core

signaling networks, further orchestrating plant growth and development in transcriptomic and metabolic levels [8,9]. Different light spectra are known to induce pronounced but differential effects on the growth and biochemical responses in vegetables [10,11].

It is widely accepted that red light has the superior quantum efficiency for photosynthesis [12], promotes seed germination [13], inversely influences the seedling morphogenesis [14], and accelerates the fruit ripening [15]. Phytochromes sensed red and far-red light, and the red to far-red light ratio is often used as a non-chemical means to regulate plant hypocotyl [16] and internode [17] elongation, leaf length and width [18], and flowering process [19]. Blue light also performs well in photosynthesis, while the high energy of this short wavelength cannot be fully utilized [20]. On other aspects, blue light participates in many critical biological processes such as phototropism [21], stomatal opening [22], chloroplast development [23,24], and leaf expansion [25]. Therefore, blue light is more considered as a growth regulator. Regarding ultraviolet radiation, plants growing outdoors are exposed to UV-A light (315–400 nm). UV-A light can trigger the phototropins and cryptochromes as blue light does but induce different responses in plants [26,27]. Generally, it leads to both inhibitory and stimulatory effects on plant photosynthesis, biomass accumulation, and morphological changes [28].

In addition to light quality, the lighting scheme is important in PFALs. It has been reported that constant spectral lighting with an unchanged light period could not maximize the growth potential of plants due to the adaptation or habits of plants to the natural light environment after the long evolution [5]. Therefore, the supplementary light mode, which simulates the change of natural light environment, begins to apply. Among all, the end-of-day light supplementary is the most typical. The EOD far-red light (17 $\mu mol \cdot m^{-2} \cdot s^{-1}$, 5 min) increased the stem length in tobacco seedling by 3.4 times comparing to EOD red light [17]. The EOD far-red light (10 $\mu mol \cdot m^{-2} \cdot s^{-1}$, 30 min) increased the internode length in poinsettia 'Christmas Spirit' and 'Christmas Eve' by 1.53 times and 1.35 times, respectively compared with EOD red light [29]. EOD blue light (50 $\mu mol \cdot m^{-2} \cdot s^{-1}$, 60 min) induced 14.29% and 17.65% higher lettuce shoot fresh mass, respectively than EOD red light and non-EOD light [5].

The plant response to light wavelength could be associated to plant characteristics such as development stages, different organs or tissues, species or cultivars, and involved with lighting methods like supplemental monochromatic wavelength or combined wavelengths, and the pre-dawn or end-of-day light supplementary. In this study, the leaf color, biomass and phytochemical contents of two lettuce cultivars were investigated under constant white light (non-EOD, control), EOD blue light, EOD far-red light, and EOD UV-A light in PFAL. The purpose of this research was to explore the application potential of different EOD lightings in the high-quality as well as high production of lettuce in PFAL. This research suggested that EOD far-red light greatly benefited the production of two lettuce cultivars, and EOD UV-A light increased the health-promoting compounds in 'Green butter' without an adverse effect on its growth.

## 2. Materials and Methods

### 2.1. Plant Materials

This experiment was conducted in the plant factory with artificial light, South China Agricultural University (East longitude 113.36°, north latitude 23.16°). Lettuce (*Lactuca sativa* L. cv 'Red butter' and 'Green butter') seeds were sown into sponge block (2 cm × 2 cm × 2 cm) with 1/4 strength Hogland nutrient solution. The full-strength nutrient solution was composed as follows: 944 mg·L$^{-1}$ Ca(NO$_3$)$_2$·4H$_2$O; 404 mg·L$^{-1}$ KNO$_3$; 160 mg·L$^{-1}$ NH$_4$NO$_3$; 200 mg·L$^{-1}$ KH$_2$PO$_4$; 348 mg·L$^{-1}$ K$_2$SO$_4$; 492 mg·L$^{-1}$ MgSO$_4$·7H$_2$O, EC ≈ 1.2 mS·cm$^{-1}$ and pH ≈ 6.4. The nutrient solution was re-circulated automatically for 10 min every half an hour. The lettuce seedlings with three expended true leaves were transplanted into the planting plate (90 cm × 60 cm, 24 plants/plate) with 1/2 strength of the nutrient solution. The environment was as follow: 22 ± 2 °C temperature, 55 ± 5% relative

humidity, $500 \pm 100$ μmol·mol$^{-1}$ $CO_2$ concentration. During the cultivation, there were rare pest or disease problems.

### 2.2. Light Treatments

The total photosynthetic photon flux density (PPFD) was 250 μmol·m$^{-2}$·s$^{-1}$, and the light period was 10 h·d$^{-1}$ (8:00–18:00). There were four lighting treatments: the white light (W) as the control and the basil light of EOD lightings, the EOD-enhanced blue light (EOD-B), the EOD supplementary far-red light (EOD-FR), and the EOD supplementary ultraviolet-A light (EOD-UV). The spectrograms and lighting patterns are presented in Figure 1, and the lighting parameters are shown in Table 1.

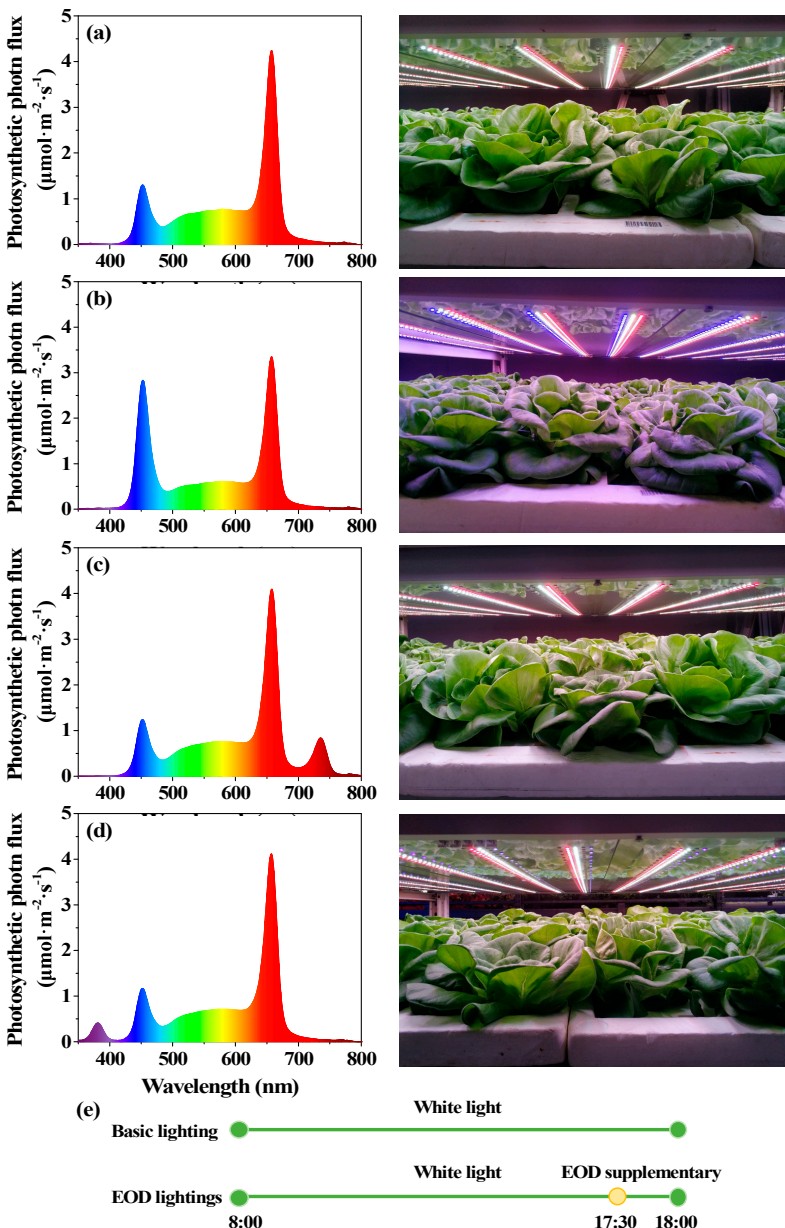

**Figure 1.** The spectrograms delivered by LEDs. Spectral distributions of (**a**) day-white light (W), (**b**) end-of-day enhanced blue light (EOD-B), (**c**) end-of-day far-red light (EOD-FR), and (**d**) end-of-day ultraviolet-A light (EOD-UV); (**e**) Patterns of end-of-day lighting treatments.

**Table 1.** Lighting parameters.

| Parameters | Lighting Treatments | | | |
|---|---|---|---|---|
| | W | EOD-B | EOD-FR | EOD-UV |
| Single-band photon flux density ($\mu$mol·m$^{-2}$·s$^{-1}$) | | | | |
| Ultraviolet light (350–400 nm) | 0.24 | 0.36 | 0.20 | 4.95 |
| Blue light (400–500 nm) | 46.12 | 83.70 | 43.72 | 41.65 |
| Green light (500–600 nm) | 69.46 | 55.09 | 68.58 | 64.34 |
| Red light (600–700 nm) | 137.38 | 107.84 | 137.03 | 135.19 |
| Far-red light (700–800 nm) | 4.79 | 3.93 | 25.99 | 5.91 |
| Integrated photon flux density ($\mu$mol·m$^{-2}$·s$^{-1}$) | | | | |
| PPFD | 252.97 | 246.54 | 249.34 | 241.18 |
| YPFD | 222.54 | 209.94 | 219.59 | 212.94 |
| TPFD | 223.63 | 210.93 | 223.78 | 216.63 |
| Radiation ratio | | | | |
| Red/Blue | 2.98 | 1.29 | 3.13 | 3.25 |
| Red/Green | 1.98 | 1.96 | 2.00 | 2.10 |
| Red/Far-red | 28.70 | 27.42 | 5.27 | 22.87 |
| Daily light integral (mol·m$^{-2}$·d) | | | | |
| 10 h | 9.11 | 8.88 | 8.98 | 8.68 |

PPFD = the photosynthetic photon flux density, YPFD = the yield photon flux density, TPFD = the total photon flux density. W= day-white light, EOD-B = end-of-day enhanced blue light, EOD-FR = end-of-day far-red light, EOD-UV = end-of-day ultraviolet-A light.

*2.3. Growth Measurements*

Sample collecting and growth measurements were carried out at 24 days after light treatments (Figure 2). Five lettuce plants in each treatment were randomly selected and weighed by an analytical balance immediately after harvest. Then the samples were oven dried at 75 °C for 72 h to determine the dry weight and calculate the moisture content.

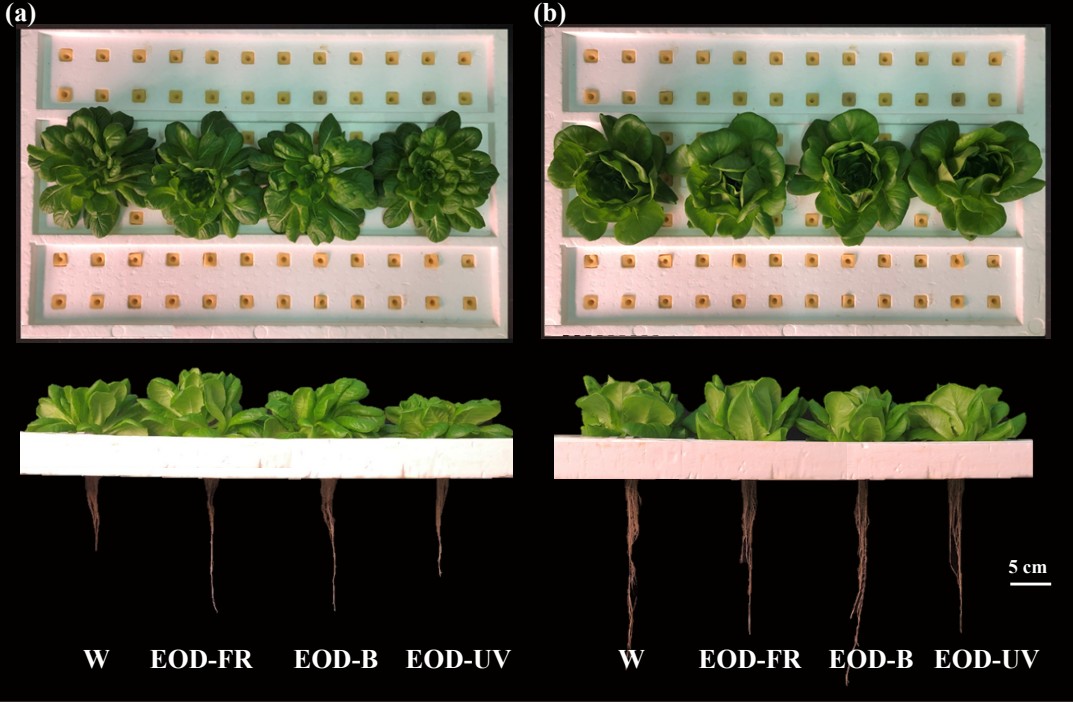

**Figure 2.** Lettuce morphology 24 days after treatments. (**a**) 'Red butter' and (**b**) 'Green butter' grown under different lighting treatments. W= day-white light, EOD-B = end-of-day enhanced blue light, EOD-FR = end-of-day far-red light, EOD-UV = end-of-day ultraviolet-A light.

## 2.4. Color Measurements

The leaf color of lettuce was non-destructively determined using a colorimeter (CR-10 plus, Konica Minolta Inc., Tokyo, Japan) at 3, 6, 9, 12, 15, 18, 21, and 24 days after light treatments (D3–D24), respectively. The value L* means lightness; a* represents the color from green to red; b* suggests the color from blue to yellow.

## 2.5. Chlorophyll and Carotenoids Measurements

The contents of chlorophyll and carotenoids were determined according to ethanol test [30]. Fresh samples of lettuce (0.2 g) were homogenized with 8 mL ethanol until the tissue turned white. The leach liquor absorbance was measured at 440 nm, 645 nm, and 663 nm by a UV-spectrophotometer (Shimadzu UV-16A, Shimadzu Corporation, Kyoto, Japan), respectively. The contents of chlorophyll and carotenoids were calculated as the following equations:

$$\text{Chlorophyll a (mg·g}^{-1}) = (12.70 \times OD_{663} - 2.69 \times OD_{645}) \times V / (1000\ W), \tag{1}$$

$$\text{Chlorophyll b (mg·g}^{-1}) = (22.88 \times OD_{645} - 4.67 \times OD_{663}) \times V / (1000\ W), \tag{2}$$

$$\text{Total chlorophylls (mg·g}^{-1}) = (8.02 \times OD_{663} + 20.20 \times OD_{645}) \times V / (1000\ W), \tag{3}$$

$$\text{Carotenoids (mg·g}^{-1}) = (4.70 \times OD_{440} - 2.17 \times OD_{663} - 5.45 \times OD_{645}) \times V / (1000\ W), \tag{4}$$

where V is the volume of extract solution (8 mL), and W is the fresh weight (0.2 g) of the sample.

## 2.6. Total Anthocyanins Measurement

The contents of total anthocyanins (TA) was measured according to pH differential method [31]. Two fresh samples of lettuce (1.0 g) were homogenized with pH 1.0 potassium chloride buffer (50 mM KCl and 150 mM HCl) and pH 4.5 sodium acetate buffer (400 mM $CH_3COONa$ and 240 mM HCl), respectively. After 5 min 14,000 g centrifuging at 4 °C, the supernatants were measured at 510 nm by a UV-spectrophotometer.

$$\text{TA (mg·g}^{-1}) = [(A_1 - A_2) \times 484.8 \times \text{dilution factor}] / 24.825, \tag{5}$$

where $A_1$ and $A_2$ are the absorbances of the sample extracted from pH 1.0 buffer and pH 4.5 buffer, respectively. The number 484.8 is the molecular weight of cyaniding-3-glucoside chloride. The number 24.825 is the absorption coefficient at 510 nm. The dilution factor in this measurement is 1.

## 2.7. Phytochemical Measurement

The DPPH radical inhibition percentage (DPPH) measurement was based on the method of Musa et al. [32]. Fresh samples of lettuce (0.5 g) were homogenized with 8 mL ethanol for 30 min in darkness. After 15 min of being centrifuged at 3000 rpm, the supernatant was used to prepare three types of mixture (Ai: Supernatant of 2 mL mixed with 2 mL 0.2 µM DPPH; Aj: Supernatant of 2 mL mixed with 2 mL ethanol; Ac: 0.2 µM DPPH mixed with 2 mL ethanol). These mixtures were determined at 517 nm by the UV-spectrophotometer. The DPPH radical inhibition percentage was calculated as follows:

$$\text{DPPH (\%)} = [1 - (Ai - Aj) / Ac] \times 100\%, \tag{6}$$

The ferric ion-reducing antioxidant power (FRAP) measurement was according to Tadolini et al. [33]. Fresh samples of lettuce (0.5 g) were homogenized with 8 mL ethanol for 30 min in darkness. After 15 min of being centrifuged at 3000 rpm, the supernatant (0.4 mL) was added to the FRAP reagent (3.7 mL), and the mixture was preserved in a 37 °C water bath for 10 min. The FRAP reagent was prepared by mixing 300 mM acetate buffer (pH 3.6), 20 mM ferric chloride, and 10 mM 2,4,6-tripyridyl-*S*-triazine (TPTZ) in 40 mM HCl in the proportion of 10:1:1 (*v:v:v*). The absorbance was then determined at

593 nm by the UV-spectrophotometer. $FeSO_4 \cdot 7H_2O$ was used as the standard, and the results were expressed as $mmol \cdot g^{-1}$ FW.

The total phenolic compounds (TPC) measurement was conducted as stated by Tadolini et al. [33]. Fresh samples of lettuce (0.5 g) were homogenized with 8 mL ethanol for 30 min in darkness. After 15 min of being centrifuged at 3000 rpm, the supernatant (1.0 mL) was mixed with 0.5 mL of Folin-ciocalteu' ultra-pure water reagent (1:1, *v:v*) and 1.5 mL 26.7% $Na_2CO_3$ solution (*w:v*). The mixture was diluted to a total volume of 10 mL with ultra-pure water. After 2 h reaction, the absorbance was recorded at 760 nm with the UV-spectrophotometer. TPC values were calculated from the gallic acid standard curve, and the results were expressed as mg gallic acid equivalent fresh weight (mg $GAE \cdot g^{-1}$ FW).

The total flavonoids (TF) measurement was in accordance with the method by Sánchez–Rangel et al. [34]. The supernatant (1 mL) was mixed with 30% ethanol (10 mL, *w:v*) and 5% $NaNO_2$ solution (0.7 mL, *w:v*). After 5 min, 10% $Al(NO_3)_3$ solution (0.7 mL, *w:v*) was added in for 5 min reaction. Then, 5% NaOH solution (5 mL, *w:v*) and 30% ethanol (8.6 mL, *v:v*) were added. The absorbance was determined at 510 nm with the UV-spectrophotometer. Rutin hydrate was used as the standard, and the results were expressed as $mg \cdot g^{-1}$ FW.

The contents of soluble proteins (SP) were determined according to Blakesley and Boezi [35]. Fresh samples of lettuce (1.0 g) were ground with 8 mL distilled water. After being centrifuged (8000 rpm, 4 °C) for 10 min, the supernatant (1 mL) was mixed with Coomassie brilliant blue G-250 solution (5 mL, 0.1 $g \cdot L^{-1}$). The absorbance was measured at 595 nm by a UV-spectrophotometer.

The contents of soluble sugars (SS) measurement were performed according to Kohyama and Nishinari [36]. Fresh samples of lettuce (1.0 g) were extracted with 80% ethanol (10 mL, *v:v*) and then homogenized with activated carbon powder (10 mg) and water bath (80 °C) for 40 min. The extract was diluted to a total volume of 25 mL with 80% ethanol (*v:v*). Then the filter liquor (0.2 mL) was mixed with diluted water (0.8 mL) and of sulfuric acid anthrone reagent (5 mL). After 10 min water bath (100 °C), the absorbance at 625 nm was detected by the UV-spectrophotometer.

The contents of vitamin C (VC) were performed referring to Shyamala and Jamuna [37]. Fresh samples of lettuce (1.0 g) were soaked in 50 mL EDTA-oxalic acid solution (200 mM EDTA and 50 mM oxalic acid), then centrifuged (5000 rpm, 4 °C) for 5 min. The supernatant (10 mL) was mixed with 3% $HPO_3$ solution (1 mL, *w:v*), 5% $H_2SO_4$ (2 mL, *v:v*) and 5% $H_8MoN_2O_4$ (4 mL, *v:v*). After 15 min incubation, the absorbance was taken at 705 nm by the UV-spectrophotometer.

The contents of nitrates were determined with the method proposed by Cataldo et al. [38]. Fresh samples of lettuce (1.0 g) were homogenized with 10 mL deionized water and water bath (100 °C) for 30 min. The filtrate was diluted with deionized water to a total volume of 25 mL. Then the extract (0.1 mL) was mixed with 5% salicylic acid-$H_2SO_4$ reagent (0.4 mL, *w:v*). After 10 min incubation, 8% NaOH (9.5 mL, *w:v*) was added and the absorbance was measured at 410 nm with a UV-spectrophotometer.

## 2.8. Data Analysis

Data were analyzed by a one-way analysis of variance (ANOVA), using SPSS 25.0 software. Significance at $p < 0.05$ was performed by the Tukey's test. XLSTAT 2019 software was used for statistical computing and multivariate principal component analysis (PCA). TBtools software [39] was used for visualizing the transformed data into a cluster heatmap.

## 3. Results

### 3.1. Growth and Biomass

The agronomic traits of lettuce cultivars were governed by the genetic as well as the interaction between genotype and environment. Most of the growth and biomass parameters were significantly influenced by cultivars, EOD light treatments, and the interaction between the two factors at harvest

(Figures 3 and 4, Tables 2 and 3). The highest fresh weight, dry weight, and moisture contents of plant and root were recorded in 'Green butter' × EOD-FR, while the lowest fresh and dry mass were observed in 'Red butter' × EOD-B (Figures 3 and 4). Concerning EOD lightings, EOD-FR stimulated the biomass of two lettuce cultivars (Figure 3). In 'Red butter', the fresh weights of total plant (28.04%) and shoot (31.34%) exhibited remarkably increases under EOD-FR, and the parallel trend was found in dry weights (14.21% and 18.37%) (Figure 3a,b,d,e). Similar in 'Green butter', the fresh weights of plant (35.68%), shoot (37.09%), and root (19.95%) were significantly increased by EOD-FR, as well as dry weights of plant and shoot (31.50% and 39.89%) (Figure 3a–e). Consequently, EOD-FR significantly decreased the root–shoot ratio in two lettuce cultivars (Figure 4d). Moreover, the growth responses to EOD-B and EOD-UV varied in terms of cultivars (Tables 2 and 3). In 'Red butter', EOD-B suppressed the fresh weights of plant (15.30%) and shoot (16.66%), as well as the homologous dry weights (13.57% and 15.92%) (Figure 3a,b,d,e). However, EOD-B led to significantly higher plant fresh weight (13.15%) in 'Green butter' (Figure 3a). Regarding EOD-UV, the fresh weights of 'Red butter' plant (15.80%) and shoot (17.44%) were increased (Figure 3a,b). However, EOD-UV had insignificant effects on the biomass of 'Green butter' (Figure 3).

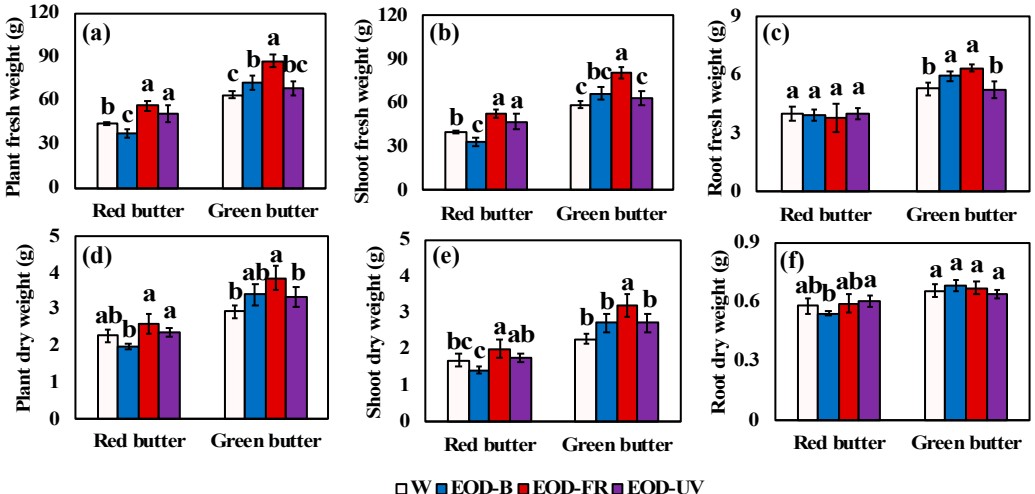

**Figure 3.** Biomass of lettuce under different end-of-day lightings. Fresh weight of plant (**a**), shoot (**b**), and root (**c**). Dry weight of plant (**d**), shoot (**e**), and root (**f**). Different letters on the top of the columns indicate significant differences at $p < 0.05$ according to one-way ANOVA, Tukey's honestly significant difference tests. W= day-white light, EOD-B = end-of-day enhanced blue light, EOD-FR = end-of-day far-red light, EOD-UV = end-of-day ultraviolet-A light.

**Table 2.** The interaction effects of cultivars and lightings on lettuce biomass.

| Interaction | Fresh Weight | | | Dry Weight | | |
|:---:|:---:|:---:|:---:|:---:|:---:|:---:|
| | **Plant** | **Shoot** | **Root** | **Plant** | **Shoot** | **Root** |
| C | *** | *** | *** | *** | *** | *** |
| L | *** | *** | NS | *** | *** | NS |
| C × L | *** | *** | * | ** | ** | ** |

NS, *, **, *** represent non-significant or significant at $p < 0.05$, 0.01, and 0.001, respectively, according to two-way ANOVA, Tukey's honest significant difference tests. C = cultivars, L = lightings.

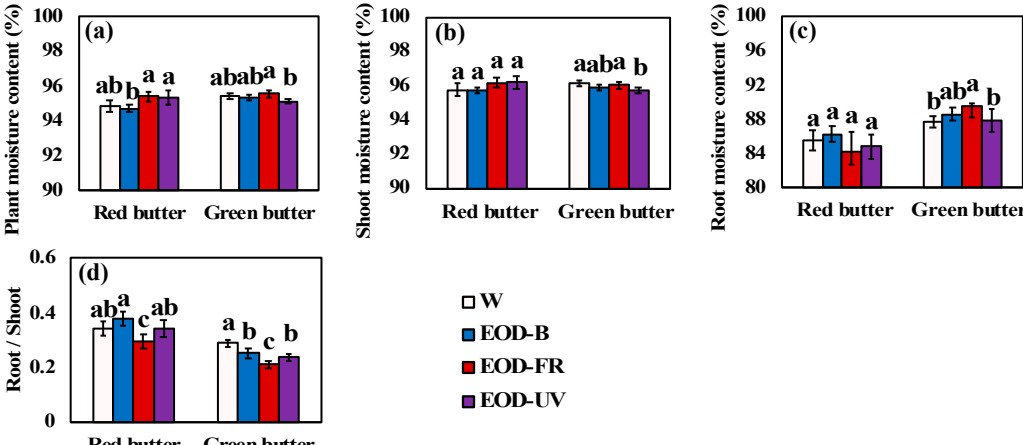

**Figure 4.** Moisture content and root–shoot ratio of lettuce under different end-of-day lightings. Moisture content of plant (**a**), shoot (**b**) and root (**c**). (**d**) was the root–shoot ratio. Different letters on the top of the columns indicate significant differences at $p < 0.05$ according to one-way ANOVA, Tukey's honest significant difference tests. W= day-white light, EOD-B = end-of-day enhanced blue light, EOD-FR = end-of-day far-red light, EOD-UV = end-of-day ultraviolet-A light.

**Table 3.** The interaction effects of cultivars and lightings on lettuce moisture content and root–shoot ratio.

| Interaction | Moisture Content | | | Root/Shoot |
|:---:|:---:|:---:|:---:|:---:|
| | **Plant** | **Shoot** | **Root** | |
| C | ** | NS | *** | *** |
| L | ** | NS | NS | *** |
| C × L | ** | ** | * | ** |

NS, *, **, *** represent non-significant or significant at $p < 0.05$, 0.01, and 0.001, respectively, according to two-way ANOVA, Tukey's honest significant difference tests. C = cultivars, L = lightings.

### 3.2. Leaf Color Transformation and Pigment Content

Two lettuce cultivars presented differential coloration during the growth (Figure 5). On the whole, 'Red butter' leaf was darker, redder, and yellower than 'Green butter' leaf. Regarding the EOD lightings, 'Red butter' had brighter, greener and more yellow leaf under all EOD lightings, while 'Green butter' leaf was darker under EOD-B and EOD-UV, and the leaf was redder and more yellow under all EOD lightings, respectively compared with W (Figure 5). In 'Red butter', the lowest redness (a* = −10.83), the highest yellowness (b* = 26.57) and lightness (L* = 39.18) were recorded at the 24th day under lighting treatment (D24), suggesting a fading coloration during growth. Whereas in 'Green butter', the highest lightness (L* = 55.26) and the deepest redness (a* = −8.42) were observed at D12 and D9, respectively.

Pigment contents and ratios were involved in lettuce leaf color. 'Red butter' possessed significantly higher contents of chlorophyll a, chlorophyll b, chlorophyll (a + b), carotenoids, and TA contents than 'Green butter' (Figure 6), characterizing two cultivars from red leaf and green leaf (Table 4). Except for carotenoids, different EOD lightings markedly influenced the pigment contents (Figure 6a–e). Interestingly, the interaction of cultivars and lightings had insignificant effects on most of the pigment contents and pigment ratios, while significant differences were observed in TA content and the ratio of chlorophyll (a + b) and TA (Table 4). In 'Red butter', TA content was greatly reduced by EOD-FR (67.41%), EOD-B (58.04%), and EOD-UV (41.07%), compared to W (Figure 6e). Consequently, the chlorophyll (a + b)/TA ratio was in the following order: EOD-FR > EOD-B > EOD-UV > W (Figure 6h). Differently in 'Green butter', the TA content was decreased by EOD-FR (83.33%) and EOD-B (81.25%) but increased by EOD-UV (30.43%) (Figure 6e). As a result, the chlorophyll (a + b)/TA ratios increased significantly under EOD-FR and EOD-B (Figure 6h).

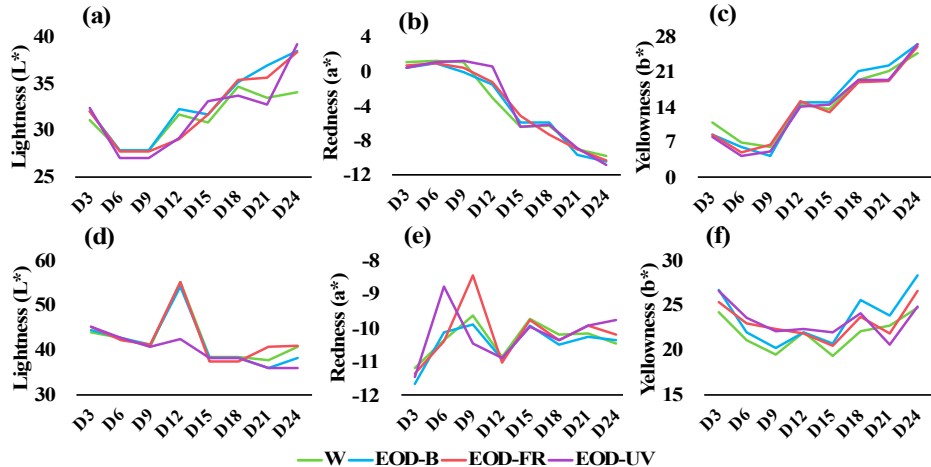

**Figure 5.** Leaf color transformation of lettuce 3 to 24 days after treatments. (**a–c**) were color parameters of 'Red butter' lettuce. (**d–f**) were color parameters of 'Green butter' lettuce. W= day-white light, EOD-B = end-of-day enhanced blue light, EOD-FR = end-of-day far-red light, EOD-UV = end-of-day ultraviolet-A light.

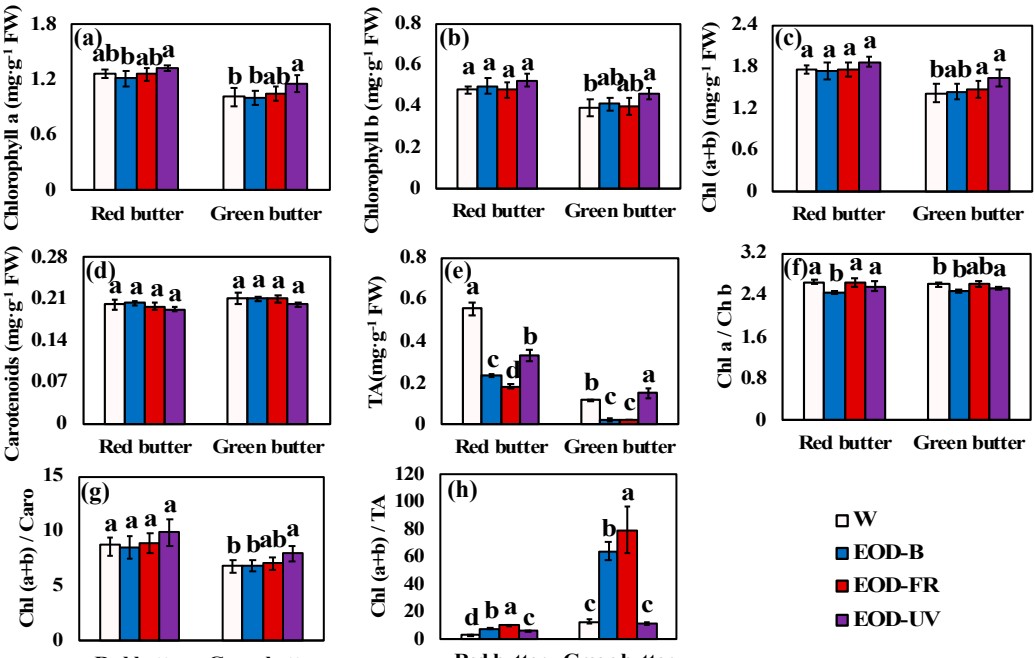

**Figure 6.** Pigment contents of lettuce under different end-of-day lightings. (**a–e**) were the pigment contents of lettuce. (**f–h**) were the pigment ratio. Different letters on the top of the columns indicate significant differences at *p* < 0.05 according to one-way ANOVA, Tukey's honest significant difference tests. W= day-white light, EOD-B = end-of-day enhanced blue light, EOD-FR = end-of-day far-red light, EOD-UV = end-of-day ultraviolet-A light. FW = fresh weight, Chl = chlorophyll, TA = total anthocyanins, Caro = carotenoids.

**Table 4.** The interaction effects of cultivars and lightings on lettuce pigments.

| Interaction | Pigments | | | | | Pigment Ratio | | |
|---|---|---|---|---|---|---|---|---|
| | Chl a | Chl b | Chl (a + b) | Caro | TA | Chl a/Chl b | Chl (a + b)/Caro | Chl (a + b)/TA |
| C | *** | *** | *** | * | *** | NS | *** | *** |
| L | ** | ** | ** | NS | *** | *** | * | *** |
| C × L | NS | NS | NS | NS | *** | NS | NS | *** |

NS, *, **, *** represent non-significant or significant at $p < 0.05$, 0.01, and 0.001, respectively, according to two-way ANOVA, Tukey's honest significant difference tests. C = cultivars, L = lightings. Chl = chlorophyll, TA = total anthocyanins, Caro = carotenoids.

### 3.3. Phytochemical Profiles

The contents of phytochemicals differed between two lettuce cultivars and among light treatments. The superior antioxidant capacity (DPPH and FRAP) and antioxidant compounds contents (TPC, TF, and VC) were observed in 'Red butter' (Figure 7 and Table 5). Concerning the interaction of cultivars and lightings, most of the antioxidant-related indices showed insignificant difference (Table 5), except that EOD-UV (17.96%) and EOD-B (16.04%) increased the DPPH compared to W in 'Green butter' (Figure 7a). EOD-FR obviously decreased the contents of SP (21.21%) and SS (27.46%) in 'Red butter', while EOD-B significantly increased SP (15.82%) and SS (6.37%) contents in 'Green butter', respectively, compared with W (Figure 7f,g). The most abundant SS content was observed in 'Green butter' × EOD-FR with the increase of 18.02%, as compared to 'Green butter' × W (Figure 7g). The lowest nitrates content was recorded in 'Red butter' × EOD-UV (0.56 mg·g$^{-1}$ FW), while the highest content was observed in 'Green butter' × EOD-UV (0.87 mg·g$^{-1}$ FW) (Figure 7h), indicating a differential accumulation of nitrates in two lettuce cultivars in response to EOD ultraviolet-A lighting.

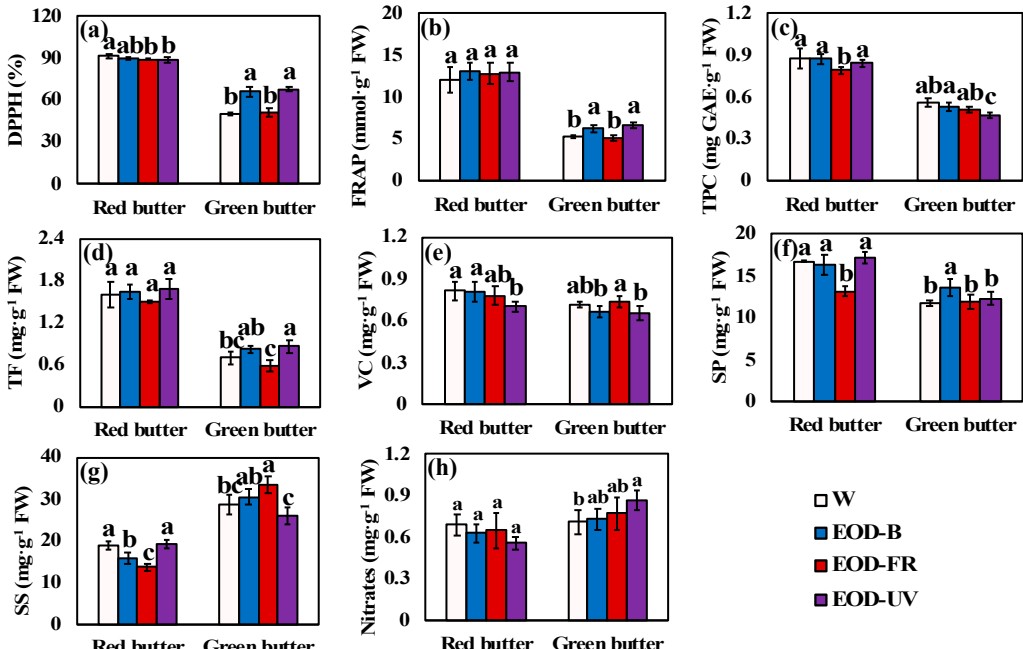

**Figure 7.** Phytochemical contents of lettuce under different end-of-day lightings. (**a**–**e**) were the contents of antioxidant compounds. (**f**–**h**) were the contents of nutrient compounds. Different letters on the top of the columns indicate significant differences at $p < 0.05$ according to one-way ANOVA, Tukey's honest significant difference tests. W= day-white light, EOD-B = end-of-day enhanced blue light, EOD-FR = end-of-day far-red light, EOD-UV = end-of-day ultraviolet-A light. FW = fresh weight, DPPH = DPPH radical inhibition percentage, FRAP = ferric ion reducing antioxidant power, TPC = total phenolic compounds, TF = total flavonoids, VC = vitamin C, SP = soluble proteins, SS = soluble sugars.

**Table 5.** The interaction effects of cultivars and lightings on lettuce phytochemical contents.

| Interaction | Antioxidant Capacity | | Antioxidant Compounds | | | Nutrient Compounds | | |
|---|---|---|---|---|---|---|---|---|
| | DPPH | FRAP | TPC | TF | VC | SP | SS | Nitrates |
| C | *** | *** | *** | *** | *** | *** | *** | *** |
| L | *** | * | ** | *** | ** | *** | NS | NS |
| C × L | *** | NS | NS | NS | NS | *** | *** | ** |

NS, *, **, *** represent non-significant or significant at $p < 0.05$, 0.01, and 0.001, respectively, according to two-way ANOVA, Tukey's honest significant difference tests. C = cultivars, L = lightings. DPPH = DPPH radical inhibition percentage, FRAP = ferric ion reducing antioxidant power, TPC = total phenolic compounds, TF = total flavonoids, VC = vitamin C, SP = soluble proteins, SS = soluble sugars.

### 3.4. Multivariate Principal Component Analysis

To compare the correlation of all growth and quality traits in two lettuce cultivars' response to different EOD lightings, the principal component analysis (PCA) was performed (Table 6 and Figure 8). The first seven principal components (F1–F6) were associated with eigen values > 1, and account for approximately 87.14% and 90.37% of the cumulative variance in 'Red butter' and 'Green butter', respectively (Table 6).

**Table 6.** Eigen value, factor scores, and contribution of the first six principal component axes to variation in lettuce under different end-of-day lightings.

| Principal Components | F1 | F2 | F3 | F4 | F5 | F6 |
|---|---|---|---|---|---|---|
| | | | *Red butter* | | | |
| Eigen Value | 7.911 | 5.18 | 3.902 | 2.709 | 1.553 | 1.402 |
| Variability (%) | 30.428 | 19.922 | 15.007 | 10.42 | 5.972 | 5.393 |
| Cumulative % | 30.428 | 50.35 | 65.358 | 75.777 | 81.749 | 87.142 |
| | | | *Green butter* | | | |
| Eigen Value | 9.814 | 6.538 | 2.848 | 1.857 | 1.249 | 1.19 |
| Variability (%) | 37.745 | 25.145 | 10.952 | 7.143 | 4.804 | 4.576 |
| Cumulative % | 37.745 | 62.89 | 73.842 | 80.985 | 85.789 | 90.365 |

F1–F6 are the first six principal component axes.

The first two factors (F1 vs. F2) of the PCA were presented in the correlation circle and scatterplot (Figure 8), and explained 50.35% of the total data variance of 'Red butter' and 62.89% for 'Green butter'. The correlation circle (Figure 8a,c) illustrated the relationships among growth parameters, antioxidants, and nutrient components, by identifying the angle between two vectors (0° < positively correlated < 90°; uncorrelated: = 90°; 90° < negatively correlated < 180°) and the distance from the center of the circle ($r > 0.5$ means relative higher correlation). From the results of 'Red butter', strong positive correlations were found between carotenoids (Caro) and nitrates contents, carotenoids and the leaf redness (a*), total flavonoids (TF) and soluble sugar (SS) contents, and among chlorophylls, respectively, while negative correlations were observed between carotenoids and the chlorophylls, carotenoids and nitrates, chlorophylls and a*, and between TPC contents and the plant and shoot fresh weights (P-FW and S-FW), respectively (Figure 8a). In 'Green butter', positive correlations were identified between nitrates and chlorophylls, among DPPH, FRAP, TF, and TA, while these indices were negatively correlated with the SS content, the leaf lightness (L*), and the leaf yellowness (b*) (Figure 8c). Interestingly, TPC contents had an insignificant correlation with the chlorophyll contents in 'Red butter', whereas a strong negative correlation in 'Green butter' was shown (Figure 8a,c). The scatterplot (Figure 8b,d) distinguished light treatments into four groups by different quadrants. The upper left quadrant of 'Red butter' and the center position of 'Green butter' sited the EOD-FR, which was characterized by higher fresh yield of lettuce plant and shoot (Figure 8b,d).

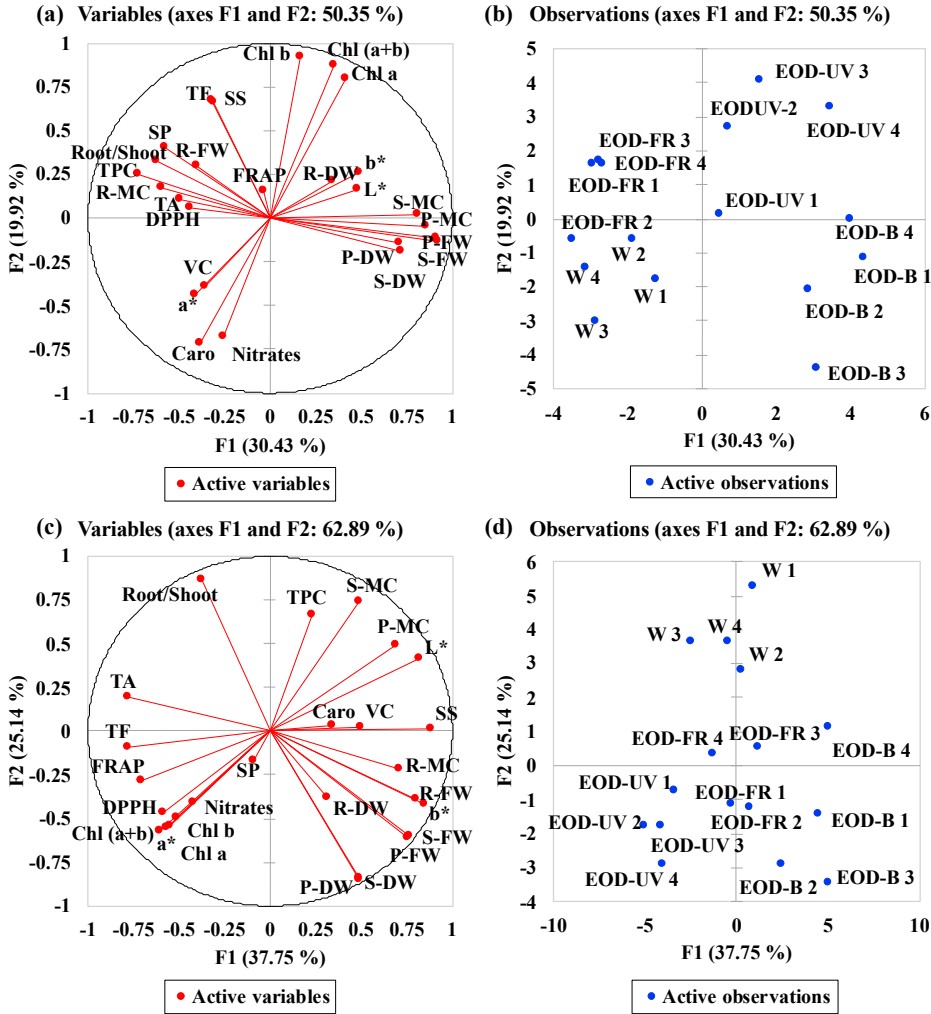

**Figure 8.** Multivariate principal component analysis showing the effects of end-of-day lightings on lettuce. (**a**) Correlation circle of 'Red butter' and (**c**) correlation circle of 'Green butter', summarizing indices relations between investigated parameters under different lighting treatments. (**b**) PCA scatter plot of 'Red butter' and (**d**) PCA scatter plot of 'Green butter', indicating distinct responses in lettuce under different lighting treatments. W= day-white light, EOD-B = end-of-day enhanced blue light, EOD-FR = end-of-day far-red light, EOD-UV = end-of-day ultraviolet-A light. FW = fresh weight, DW = dry weight, MC = moisture content, P- = plant-, S- = shoot-, R- = root-. Chl = chlorophyll, TA = total anthocyanins, Caro = carotenoids. DPPH = DPPH radical inhibition percentage, FRAP = ferric ion reducing antioxidant power, TPC = total phenolic compounds, TF = total flavonoids, VC = vitamin C, SP = soluble proteins, SS = soluble sugars.

### 3.5. Heatmap Analysis

A heatmap synthesizing the response of the measured parameters provided an integrated view of the effect of different EOD lightings on the growth and quality of lettuce (Figure 9).

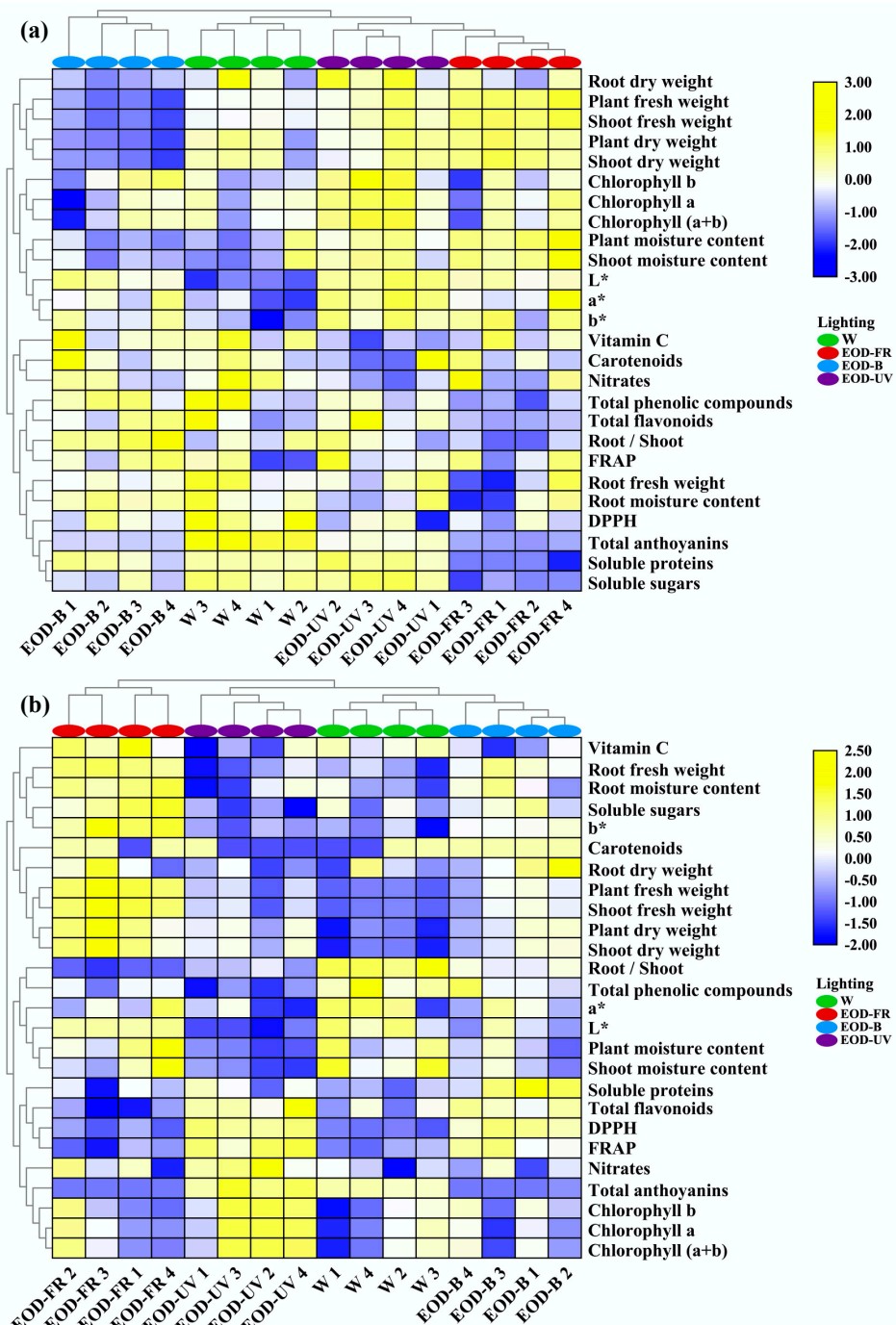

**Figure 9.** Cluster heatmap analysis summarizing lettuce responses to end-of-day lightings. (**a**) Heatmap of 'Red butter' and (**b**) 'Green butter' responses. Results are visualized using a false color scale with yellow indicating an increased parameter, and blue indicating a decreased parameter. W= day-white light, EOD-B = end-of-day enhanced blue light, EOD-FR = end-of-day far-red light, EOD-UV = end-of-day ultraviolet-A light.

Regarding 'Red butter', the EOD-UV and EOD-FR clusters were the closest to each other in terms of measured parameter responses, and they were equidistant from cluster W (Figure 9a). The EOD-UV and EOD-FR clusters both showed brighter and yellower leaf, as well as higher fresh yield and dry weight (Figure 9a). Meanwhile, cluster EOD-B was considerably separated from the other three clusters: EOD-B decreased the fresh and dry weight of plant and shoot; increased the root-shoot ratio; and led to higher contents of VC, carotenoids, nitrates, TPC, and TF compared to W, EOD-FR,

and EOD-UV, contributing to separate the EOD-B cluster from the others (Figure 9a). Moreover, the heatmap indicated an affinitive pattern in the leaf color parameters (L*, a*, and b*) and the plant and shoot water content, as well as the content of chlorophylls (Figure 9a).

However, in 'Green butter', the W and EOD-B clusters showed highly similar response patterns, and they were equidistant from cluster EOD-UV (Figure 9b). Whereas, cluster EOD-FR was separated from the other three clusters: EOD-FR elicited higher accumulations of VC, SS, carotenoids, and promoted the fresh and dry weight of lettuce (Figure 9b). At the same time, the EOD-UV cluster facilitated the accumulation of antioxidants (total anthocyanins, total flavonoids, DPPH, and FRAP) and chlorophylls (Figure 9b). Interestingly, the patterns of a* and L* in 'Green butter' were close to the plant and shoot water content, but b* was close to SS contents, which differed from 'Red butter' (Figure 9b).

## 4. Discussion

### 4.1. Lettuce Biomass in Response to End-Of-Day Lightings Growth and Biomass

'Green butter' possessed higher biomass (73.04 g/plant) than 'Red butter' (47.18 g/plant) (Table 2). Among all EOD lightings, EOD-FR showed the highest fresh weight of plant (71.65 g), shoot (66.57 g), and root (5.08 g) (Figure 3a–c). There were positive FR effects on the plant fresh weight and dry mass. FR supplementation enhanced the fresh weight (30.71%) and dry weight (14.52%) of lettuce 'Red Cross' [40] and increased the fresh mass (14.91%) of lettuce 'Outredgeous' [41]. The addition of FR (16–64 $\mu mol \cdot m^{-2} \cdot s^{-1}$) promoted the shoot dry weight (28–50%) of geranium 'Pinto Premium Orange Bicolor' and snapdragon 'Trailing Candy Showers Yellow' [42]. EOD FR (10 $\mu mol \cdot m^{-2} \cdot s^{-1}$, 30 min) benefited the poinsettia 'Christmas Spirit' in stem dry matter accumulation (100.00%), while it had an insignificant influence in the dry matter of poinsettia 'Christmas Eve' [29]. However, RGB light spectrum with 18% FR (21.6 $\mu mol \cdot m^{-2} \cdot s^{-1}$) inhibited the shoot dry weight (34.87%) of bell pepper seedlings [43] and retarded the shoot fresh weight (35.92%), root fresh weight (69.15%), shoot dry weight (36.49%), and root dry weight (72.73%) in lettuce 'Green Oak Leaf' [44]. In this study, with regard to C × L, EOD-FR performed in promoting the fresh weight of plant and shoot in both 'Red butter' (28.04% and 31.34%) and 'Green butter' (35.68% and 37.09%). These variable responses might be the interaction with species, genotypes, lighting duration and methods, and other environment factors. In addition, the decreased root–shoot ratio suggested that EOD-FR was more conducive to the shoot growth (Figure 4d).

FR supplementation is in favor of shoot growth by enlarging the leaf area and increasing the stem and petiole length, thereby facilitating light interception and ultimately eliciting better biomass production [42,45,46]. Moreover, the increased moisture content indicated that absorbing more water under EOD-FR might partially contribute to the higher fresh weight of lettuce (Figures 4 and 8). Furthermore, FR can also drive photochemistry and photosynthesis by increasing the photosynthetic rate [47], quantum yield of photosystem II, as well as the net photosynthesis [48]; FR light was positively related to the whole plant net assimilation [42,49].

Blue light and UV-A light also affected the growth and biomass of plants. Chen et al. (2016) reported that blue light (30 $\mu mol \cdot m^{-2} \cdot s^{-1}$, 16 h) increased the shoot dry weight (47.30%) of lettuce 'Green Oak Leaf'. Adding 9% blue light to red light resulted in higher fresh and dry mass in lettuce (21.03% and 76.69%), spinach (44.09% and 43.55%), kale (117.20% and 42.26%), basil (35.43% and 43.54%), and pepper fruits (10.16% and 66.33%) [50]. Similarly, 'Green butter' exhibited significantly higher plant fresh weight (13.16%) and shoot dry weight (19.30%) under EOD-B, but 'Red butter' seemed less sensitive to EOD blue light (Table 2). In cucumber plants 'Hi Jack', the stem weight pre unit of stem length (29%), leaf dry weight (16%), and root dry weight (47%) were larger under UV-A-enriched radiation (3.6 $W \cdot m^{-2}$, 4 h) [51]. In lettuce 'Klee', UV-A light (10–30 $\mu mol \cdot m^{-2} \cdot s^{-1}$, 16 h) enhanced the shoot fresh weight (15–31%) and shoot dry weight (15–29%) [24]. Whereas, EOD-UV diminished the shoot dry weight (19.30%) in 'Green butter' and had no significant effects on fresh and

dry mass of 'Red butter' (Figure 3). These might be due to the weaker and shorter EOD ultraviolet-A light in this study.

### 4.2. Leaf Color Responds to End-Of-Day Lightings

The additional FR radiation decreased the chlorophyll concentration per unit leaf area [42] and reduced the amount of chlorophyll (14%) and carotenoids (11%) [44] in lettuce. The tea (cv. Hangjinya) leaf under higher blue light ratio possessed higher b* value (yellower), which might be due to the lower contents of chlorophyll a, chlorophyll b, and chlorophyll (a + b) [52]. UV-A radiation elicited the anthocyanins accumulation in the hypocotyls in soybean sprouts, and the trend was consistent with the expression pattern of anthocyanin biosynthesis-related genes [53].

No similarity was found in this study; the color of lettuce roughly followed a similar change process throughout the growth stage, which presented a decrease in redness and an increase in yellowness (Figure 5). The reason might be related to the dynamically varied relationship between the lettuce growth rate and pigment levels. The lightness (L*) of 'Red butter' increased from D6 to D24, and the highest value was recorded under EOD-UV at D24 (Figure 5a). From D9 to D 24, the redness (a*) of 'Red butter' leaf gradually faded while the yellowness (b*) increased (Figure 5b,c). However, a* and b* values of 'Red butter' seemed to be unaffected by EOD lightings (Figure 5b,c). Whereas, the leaf color of 'Green butter' exhibited fluctuation during the growth time (Figure 5d–f). The L* value under EOD-UV remained stable throughout the growth period, while the peak under other lightings was observed at D12 (Figure 5d). The peak of a* value was observed at D6 × EOD-UV and D9 × EOD-FR, which might be caused by the pigment biosynthesis in adaption to EOD UV-A and FR lights at early growth stage (Figure 5e). However, the color parameters of 'Green butter' at D24 were not significantly different among EOD lightings (Figure 3d–f). These results were consistent with the unchanged contents of chlorophylls and carotenoids (Figure 6a–d). Moreover, compared with chlorophylls contents (1.78 and 1.49 mg·g$^{-1}$), the TA contents (0.33 and 0.08 mg·g$^{-1}$) were not dominant either in 'Red butter' or 'Green butter' (Figure 6c,e). Therefore, the effects of significantly decreased TA content under EOD lightings could not be reflected in color parameters.

To conclude, EOD lightings of blue light, far-red light, and ultraviolet-A light could not significantly affect the leaf color of lettuce. The effects of EOD lightings might be to moderately aggravate/weaken the degree of color change or to accelerate the change trend, but it cannot reverse or eliminate these changes.

### 4.3. Lettuce Phytochemical Profiles in Relation to End-Of-Day Lightings

Plant species and cultivars respond to light recipes in different ways, and genotypic effect is the principle quantitative and qualitative variation in vegetables metabolites contents [54]. Different light recipes can lead to remarkable changes in plant transcriptomic pathways, but the metabolic traits may behave differently in different genotypes [55]. 'Red butter' presented stronger antioxidant capacity (DPPH and FRAP) and higher contents of antioxidant compounds (TPC, TF, and VC) than 'Green butter' (Figure 7a–e). With respect to C × L, significant enhancement of DPPH was observed under EOD-B and EOD-UV in 'Green butter' (Figure 7a and Table 5). The DPPH increased more in lettuce (1.3 times), spinach (1.2 times), and kale (1.2 times) under 17% added blue light than 100% red light, while it was enhanced in basil (1.2 times) and sweet pepper (1.1 times) under 9% added blue light [50]. Whereas, the antioxidant-related metabolites in 'Red butter' were little affected by EOD lightings (Table 5). Analogously, neither supplemental FR (160.4 μmol·m$^{-2}$·s$^{-1}$, 16 h) nor UV-A (20.9 μmol·m$^{-2}$·s$^{-1}$, 16 h) affected the contents of phenolics and ascorbic acid in 'Red Cross' lettuce [40]. UV light is often regarded as an abiotic stress to plant that stimulates the accumulation of reactive oxygen species in plants [56] and activates the defense and disease-resistance mechanisms [57]. In 'Klee' lettuce, the antioxidant contents of total phenolic (17.78%), total flavonoids (48.33%), and ascorbic acid (61.04%) were greatly simulated under UV-A supplementation (30 μmol·m$^{-2}$·s$^{-1}$, 16 h) [24]. The increased DPPH in lettuce might be a response to the slight stress caused by EOD-UV and EOD-B. On the contrary, UV solar exclusion (exclusion of more than 99% of UV-A) led to significantly higher

DPPH (17.16%) and total phenolics content (51.54%) in the spicas of *Prunella vulgaris* L. than under solar control [58].

Soluble sugar content was greatly reduced by 27.46% in 'Red butter' × EOD-FR, while it increased by 16.76% in 'Green butter' × EOD-FR (Figure 7g). Compared to the corresponding C × W, the soluble proteins content was significantly lower in 'Red butter' × EOD-FR (21.21%), while it was higher in 'Green butter' × EOD-B (15.82%) (Figure 7f). Differential sugar and protein contents were also observed in other lettuce cultivars in response to UV-A lights. Supplemental UV-A (10, 20, and 30 $\mu$mol·m$^{-2}$·s$^{-1}$, 16 h) stimulated the soluble sugar content (12.74–26.11%) and total soluble protein content (13.73–23.53%) in 'Klee' lettuce, and the 10 $\mu$mol·m$^{-2}$·s$^{-1}$ UV-A obtained the best promotion effects [24]. Supplemental UV-A light (6 $\mu$mol·m$^{-2}$·s$^{-1}$, 16 h) resulted in 1.76 times higher maltose content and nearly no change in total proteins in red leaf 'Red cos' lettuce, but it had no effects on maltose and total proteins in 'Lobjoits green cos' lettuce [55]. These suggested that the synthesis and/or metabolic processes of sugar and proteins in response to end-of-day FR, B, and UV-A were different in two lettuce cultivars.

Nitrate was the main form of nitrogen uptake in plants; there was, directly or indirectly, a transformation among sugars, proteins and nitrates concerning the ratio of carbon and nitrogen [59]. Nitrates content was lower in 'Red butter' (0.63 mg·g$^{-1}$) than 'Green butter' (0.77 mg·g$^{-1}$). The UV exclusion was reported to cause a significant debility in nitrate reductase activity, retarding the catalyze process that nitrate transformed into nitrite [60]. However, an increased nitrates content was observed in 'Green butter' × EOD-UV (Figure 7h), suggesting a distinct responses of plant nitrate contents in response to day-UV light and EOD-UV light.

Both PCA analysis and heatmap analysis validated the differential growth and phytochemical profiles of two lettuce cultivars under different EOD lightings (Figures 4 and 5). There were strong positive correlations among the a* value, carotenoids content, and nitrate contents in 'Red butter'; and these three indices were negatively related to the contents of chlorophyll a, chlorophyll b, chlorophyll (a + b), b* value, and L* value (Figure 8a). Analogously in 'Green butter', the L* value was negative related to a* value, but it had insignificant correlation with carotenoids (Figure 8c). Although PCA described the correlations among different parameters, the characteristics of each EOD lighting could not be represented visually. Thus, we used the heatmap to provide a global view of agronomic and metabolic traits and identify the phenotypic variation patterns associated with different EOD lightings (Figure 9). From the heatmap, similar and differential response patterns of two lettuce cultivars under EOD lightings were observed. In 'Red butter', the comparison of EOD-FR vs. EOD-UV had similar characteristics (Figure 9a). However, in 'Green butter', the close patterns were changed to the comparison of W vs. EOD-B (Figure 9b). Overall, these results verified that different EOD light treatments evoke specific growth and metabolic responses in 'Red butter' and 'Green butter'.

## 5. Conclusions

In this paper, we investigated the potential of end-of-day blue, far-red, and ultraviolet-A light to regulate the leaf color, growth and phytochemical accumulation of two lettuce cultivars. 'Green butter' showed higher fresh yield, while 'Red butter' possessed higher antioxidant and nutrient values. EOD far-red light performed by increasing plant and shoot fresh yield in two lettuce cultivars, while slightly decreasing the nutrition compounds. The application of EOD blue light and EOD UV-A light in lettuce required specific cultivars consideration.

Considering the low intensity and short duration of EOD light supplementation used in this study, further researches are needed, including different light intensity, supplemental periods, and switches of EOD light qualities. Moreover, the photosynthesis parameters, related enzyme contents, and gene expression patterns can lead to a better understanding of the signal and energy effects of EOD lightings.

**Author Contributions:** Y.L., R.S., H.J. and L.W. carried out the experiments. S.S., Y.Z., W.S. and H.L. performed, analyzed, and/or supervised this work. Y.L. and H.L. helped with data analysis and drafted the manuscript. H.L. acquired of funding and helped to draft the manuscript. All authors have read and agreed to the published version of the manuscript.

**Funding:** This work was supported by Key Research and Development Program of Guangdong (2019B020214005, 2019B020222003).

**Conflicts of Interest:** The authors declare no conflict of interest.

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
