# Peer review of "End-Of-Day LED Lightings Influence the Leaf Color, Growth and Phytochemicals in Two Cultivars of Lettuce"

_agronomy, doi:10.3390/agronomy10101475_

Round 1
Reviewer 1 Report
The current manuscript has discussed in detail about using end-of-day lighting which has significant effects on the lettuces. The experimental design is reasonable and understandable. even though the corresponded author has just published a paper "Growth, Nutritional Quality and Health-Promoting Compounds in Chinese Kale Grown under Different Ratios of Red:Blue LED Lights" in the Agronomy journal. One minor suggestion is that the abbreviation of the name of light could be changed.
W: white light; WB: end-of-day enhanced blue light; WFR: end-of13 day supplementary far-red light; WUV: end-of-day supplementary ultraviolet-A light
Other comments:
1. PFAL should be the abbreviation of plant factory using artificial light
2. In the end of introduction, line 78, the author should describe which light can help the production of lettuce and which light can help the growth of lettuce with high quality.
3. In the section 3.2 Leaf Color Transformation and Pigment Content, the manuscript only shows the results without the discussion. Why the redness go down, while the yellowness go up?
4. All the tables are suggested to use histogram or bar graph.
Author Response
Sep. 18, 2020
Dear Editors,
Thanks for the kind comments and valuable suggestions from the reviewers and yours. We have carefully revised the manuscript (agronomy-929664) according to these suggestions. All the revisions were clearly highlighted in red.
Enclosed, please find the revised manuscript and replies to the reviewers’ comments. We hope that our effort has made this manuscript better. Thank you for providing us the opportunity to revise our manuscript and we look forward to having your editorial decision.
Best wishes.
Yours sincerely,
Pro. Dr. Houcheng Liu
College of Horticulture
South China Agricultural University
Guangzhou, 501642
P.R. China
Tel: +86-13380055767
Email: liuhch@scau.edu.cn
Replies to the Reviewer1's comments
Thank you for your recognition on our research. This manuscript has been revised according to your comments and suggestions. All the revisions were clearly highlighted in red. The replies to your comments are listed as follows:
- One minor suggestion is that the abbreviation of the name of light could be changed. (W: white light; WB: end-of-day enhanced blue light; WFR: end-of day supplementary far-red light; WUV: end-of-day supplementary ultraviolet-A light)
Answers: The abbreviations of the name of light have been changed into W, EOD-B, EOD-FR, and EOD-UV. Please see them in line 12-13.
- PFAL should be the abbreviation of plant factory using artificial light.
Answers: The full name of PFALs has been replaced by the plant factories with artificial light (PFALs). Please find the revise in line 34-35.
- In the end of introduction, line 78, the author should describe which light can help the production of lettuce and which light can help the growth of lettuce with high quality.
Answers: The description of superior light in increasing the lettuce production and quality has been added in the end of introduction. Please find the revise in line 79-81.
- In the section 3.2 Leaf Color Transformation and Pigment Content, the manuscript only shows the results without the discussion. Why the redness go down, while the yellowness go up?
Answers: As can be seen from the overall trend of the line chart (Figure 5), the color of lettuce roughly follows a similar change process throughout the growth stage -- the redness decreases and the yellowness increases. Since we did not perform dynamic destructive sampling, we can only suppose that the reason might be related to the dynamically varied relationship between the lettuce growth rate and pigment levels. In this process, the effects of EOD light treatments might be to aggravate/weaken the degree of color change (e.g. Figure 5d, f) or to accelerate the change of color (e.g. Figure 5e), but it cannot reverse or eliminate these changes. The discussion was revised as above, please find it in 4.2 Leaf Color Responds to End-of-day Lightings, line 411-414 and 428-431.
- All the tables are suggested to use histogram or bar graph.
Answers: Table 2-5 have been adjusted to bar graphs (Figure 3-7), and the interaction of cultivars and lightings on lettuce were presented as tables (Table 2-5). The tables and figures cited in the manuscript were revised accordingly. Please find them in the manuscript.

Reviewer 2 Report
Li et al reported a study regarding the end-of-day led lightings Influence the leaf color, growth and phytochemicals in two cultivars of lettuce
The manuscript is well written and easy to read. The experiment is well organized and results well-presented and discussed.
I have only few minor suggestions:
I suggest to include a picture of the experiment
The description of the agronomic management should be improved: any pest management? How the nutrient solution was managed? How many time replaced?
I suggest to include the description of the place where the experiment was performed. In greenhouse? If yes, please report the description of the greenhouse
In the caption of tables and figures should be reported all the abbreviations displayed in the corresponding table/figure
Author Response
Sep. 18, 2020
Dear Editors,
Thanks for the kind comments and valuable suggestions from the reviewers and yours. We have carefully revised the manuscript (agronomy-929664) according to these suggestions. All the revisions were clearly highlighted in red.
Enclosed, please find the revised manuscript and replies to the reviewers’ comments. We hope that our effort has made this manuscript better. Thank you for providing us the opportunity to revise our manuscript and we look forward to having your editorial decision.
Best wishes.
Yours sincerely,
Pro. Dr. Houcheng Liu
College of Horticulture
South China Agricultural University
Guangzhou, 510642
P.R. China
Tel: +86-13380055767
Email: liuhch@scau.edu.cn
Replies to the Reviewer2's comments
Thank you for your time and efforts on our manuscript. This manuscript has been revised according to your comments and suggestions. All the revisions were clearly highlighted in red. The replies to your comments are listed as follows:
- I suggest to include a picture of the experiment.
Answers: The experiment picture was added as a part of Figure 1. Please see it in line 102.
- The description of the agronomic management should be improved: any pest management? How the nutrient solution was managed? How many time replaced?
Answers:
(1) Different from open-land cultivation, the lettuce plants were cultivated in the plant factory with strict isolation and environmental controls (fresh air circulation, air conditioner, air and water filtration, etc). Therefore, there were rare pest or disease problems. Please find the revise in line 93-94.
(2) The nutrient solution was re-circulated automatically from supply tank (3 tones in total) for 10 minutes every half an hour, ensuring the balance of nutrients and sufficient dissolved oxygen. Please find the revise in line 89-90.
(3) During the 24 days of cultivation, the pH and EC of the nutrient solution were stable, and the lettuce roots grown healthily. According to our cultivation experience, water conservation and environmental protection needs, we supposed it was not necessary to renew the nutrient solution.
- I suggest to include the description of the place where the experiment was performed. In greenhouse? If yes, please report the description of the greenhouse.
Answers: In line 84-85, we mentioned that the experiment was performed in the plant factory with artificial light. The environment parameters of the plant factory can be seen in line 92-93.
- In the caption of tables and figures should be reported all the abbreviations displayed in the corresponding table/figure.
Answers: The full name of all the abbreviations have been added in the caption of corresponding table/figure. Please find them in the manuscript.
